# MICA+ Tumor Cell Upregulated Macrophage-Secreted MMP9 via PROS1-AXL Axis to Induce Tumor Immune Escape in Advanced Hepatocellular Carcinoma (HCC)

**DOI:** 10.3390/cancers16020269

**Published:** 2024-01-08

**Authors:** Qiulin Wu, Xicai Li, Yan Yang, Jingquan Huang, Ming Yao, Jianjun Li, Yubin Huang, Xiaoyong Cai, David A. Geller, Yihe Yan

**Affiliations:** 1Department of General Surgery, The Second Affiliated Hospital of Guangxi Medical University, Nanning 530007, China; wuqiulin2020@163.com (Q.W.); lixicai20213@163.com (X.L.); yangyanmail0804@163.com (Y.Y.); hjq08287712@163.com (J.H.); yaoming20130703@163.com (M.Y.); lijianjunmail@163.com (J.L.); efy9916@126.com (Y.H.); cxy0771@163.com (X.C.); 2Thomas E. Starzl Transplantation Institute, Department of Surgery, University of Pittsburgh Medical Center, Pittsburgh, PA 15260, USA

**Keywords:** HCC, MICA, PROS1, AXL, MMP9, macrophage

## Abstract

**Simple Summary:**

The effect of macrophages on tumor cells in the tumor microenvironment of hepatocellular carcinoma (HCC) has attracted more attention. In this study, expression levels of MMP9 and MICA were significantly elevated in macrophages and tumor cells, respectively. The IFN-γ pathway was activated in MICA+ tumor cells and MMP9+ macrophages. The interaction between MICA+ tumor cells and MMP9+ macrophages was mediated through the PROS1-AXL pathway. MMP9 was mainly expressed in M2-like macrophages. IRF1 induced the upregulation of PROS1 and MICA in HCC cells. MICA+ tumor cells stimulated the secretion of MMP9 from macrophages through the PROS1-AXL axis, thereby facilitating the proteolytic shedding of MICA into soluble MICA. This study provided valuable insights and supported the therapeutic application of AXL inhibition in HCC.

**Abstract:**

Background: tumor-associated macrophages (TAMs) constitute a significant proportion of non-cancerous cells within the intricate tumor microenvironment (TME) of hepatocellular carcinoma (HCC). Understanding the communication between macrophages and tumor cells, as well as investigating potential signaling pathways, holds promise for enhancing therapeutic responses in HCC. Methods: single-cell RNA-sequencing data and bulk RNA-sequencing data were derived from open source databases Gene Expression Omnibus (GEO) and The Cancer Genome Atlas (TCGA). Through this analysis, we elucidated the interactions between MICA+ tumor cells and MMP9+ macrophages, primarily mediated via the PROS1-AXL axis in advanced HCC. Subsequently, we employed a range of experimental techniques including lentivirus infection, recombinant protein stimulation, and AXL inhibition experiments to validate these interactions and unravel the underlying mechanisms. Results: we presented a single-cell atlas of advanced HCC, highlighting the expression patterns of MICA and MMP9 in tumor cells and macrophages, respectively. Activation of the interferon gamma (IFN-γ) signaling pathway was observed in MICA+ tumor cells and MMP9+ macrophages. We identified the existence of an interaction between MICA+ tumor cells and MMP9+ macrophages mediated via the PROS1-AXL axis. Additionally, we found MMP9+ macrophages had a positive correlation with M2-like macrophages. Subsequently, experiments validated that DNA damage not only induced MICA expression in tumor cells via IRF1, but also upregulated PROS1 levels in HCC cells, stimulating macrophages to secrete MMP9. Consequently, MMP9 led to the proteolysis of MICA. Conclusion: MICA+ HCC cells secreted PROS1, which upregulated MMP9 expression in macrophages through AXL receptors. The increased MMP9 activity resulted in the proteolytic shedding of MICA, leading to the release of soluble MICA (sMICA) and the subsequent facilitation of tumor immune escape.

## 1. Introduction

Hepatocellular carcinoma (HCC) is the predominant form of liver cancer, accounting for 80–90% of cases [1]. It is a significant global health concern, particularly in countries such as China [2]. While early-stage HCC can be effectively treated with surgical resection, ablation, or liver transplantation, the majority of patients are diagnosed at advanced stages, resulting in poor prognosis and limited treatment options [3]. Despite advancements in systemic therapies including immunotherapy, molecular targeted therapy, and chemotherapy, the complex tumor microenvironment (TME) of HCC poses challenges to treatment efficacy [3,4]. Within the TME, tumor-associated macrophages (TAMs) contribute to various tumor-promoting processes, such as immune suppression, metastasis, angiogenesis, maintenance of cancer cell stemness, and therapeutic resistance [5]. Consequently, it is imperative to elucidate the molecular mechanisms by which TAMs contribute to treatment resistance in HCC.

Our previous investigations have revealed the pivotal role of interferon regulatory factor 1 (IRF1) as a central transcription factor in the interferon (IFN) pathway, exerting significant influence on cell proliferation, apoptosis, tumorigenesis, and the TME [6,7,8]. Subsequent study demonstrated that DNA damage response induced by chemotherapy and molecular targeted therapy upregulated the expression of major histocompatibility complex (MHC) class I associated chains A (MICA) in HCC cells through IRF1 mediation [9], and increased the number of natural killer (NK) cells and CD8+ T cells through the activation of natural killer group 2D (NKG2D) receptors [10]. Conversely, matrix metalloproteinases (MMPs) and a disintegrin and metalloproteinases (ADAMs) secreted by tumor cells or non-cancerous cells within the TME have been implicated in the proteolytic shedding of MICA [11]. However, the mechanisms underlying the release of soluble MICA (sMICA) from cancer cell membranes remained poorly understood [12].

Matrix metalloproteinase 9 (MMP9) possesses the ability to hydrolyze various structural and signaling proteins, and is considered a potential inhibitor of immune response and immune cell transport. The elevation of MMP9 was associated with tumor growth and metastasis promotion in some cancers [13]. Wang et al. found that the expression of MMP9 was positively correlated with the infiltration of Th1 cells, T follicular helper cells, neutrophils, and macrophages [14]. Moreover, the expression of MMP9 was predominantly observed in monocytes rather than hepatoma cells [15]. MMP9-positive macrophages have been shown to play crucial roles in tumor tissue remodeling, monocyte migration, tumor cell migration and metastasis, as well as tumor immune evasion and response to immunotherapy [15].

Protein S 1 (PROS1) has been identified as a ligand for the Tyro3, AXL, and Mer (TAM) family of tyrosine kinase receptors [16], making it a potential target for various cancer types [17]. Additionally, AXL has been implicated in innate or acquired resistance to chemotherapy, molecular targeted therapy, and radiotherapy [18]. Furthermore, the AXL signaling pathway not only contributes to the progression and resistance phenotypes of HCC [19], but also exerts suppressive effects on both adaptive and innate immune responses against malignancies [20,21]. However, the precise role of the PROS1-AXL axis in regulating MMP9 secretion from macrophages and its consequent impact on the proteolytic shedding of MICA remains incompletely understood.

In this study, we presented a single-cell atlas of advanced HCC, revealing distinct expression patterns of MICA and MMP9 in tumor cells and macrophages, respectively. Furthermore, it was observed that MICA-positive HCC cells induced MMP9 secretion from macrophages through the PROS1-AXL axis. This highlighted that MMP9 secretion subsequently facilitated the proteolysis of MICA, thereby promoting the evasion of the tumor from immune surveillance. These groundbreaking discoveries revealed the mechanism enabling the evasion of NKG2D-mediated immune responses.

## 2. Materials and Methods

### 2.1. Acquirement of scRNA-Seq and Bulk-RNA-Seq Data

In order to reduce the influence of different hepatitis backgrounds and tumor stages on the cellular components in TME, the scRNA-seq data of 4 samples (GSM4505960, GSM4505961, GSM4505963 and GSM4505964) of GSE149614 [15] were downloaded from the Gene Expression Omnibus (GEO) database (https://www.ncbi.nlm.nih.gov/geo (accessed on 13 August 2022)). A total of 10,649 cells and 25,712 genes from two patients including two tumor samples and two adjacent non-cancerous samples data were obtained [15], comprising 5,615 cells from tumor tissue and 5034 cells from non-tumor tissue. The details on gene numbers (nFeature_RNA), sequencing depth (nCount_RNA), and the percentage of mitochondrial genes (percent.mt) were demonstrated (Appendix A). The Pearson correlation coefficient between gene count and sequencing depth was determined to be 0.92 for tumor samples and 0.71 for normal samples (Appendix A), indicating a positive correlation. Quality control measures were implemented for the single-cell transcriptome data, including a feature gene count ranging from 200 to 6000 and a threshold of less than 15% for mitochondrial proportion (percent.mt) for data screening and filtering. Consequently, a total of 10,338 cells (5359 tumor cells and 4979 normal cells) were included in subsequent analyses. The bulk RNA-sequencing data and corresponding clinical information of HCC patients were downloaded from The Cancer Genome Atlas (TCGA) database (https://portal.gdc.cancer.gov (accessed on 20 August 2022)) containing 50 normal samples and 374 tumor samples. After excluding tumor patients without survival data, 365 tumor patients were maintained in this analysis.

### 2.2. Patient Samples

All tumor and adjacent liver tissues were captured from patients at the Department of General Surgery, the Second Affiliated Hospital of Guangxi Medical University (Nanning, China). This study was approved by the Second Affiliated Hospital Ethics Committee (No. 2022-KY-0208) and conducted in accordance with the Declaration of Helsinki.

### 2.3. scRNA-Seq Data Underwent Preprocessing and Analysis

The scRNA-seq data underwent preprocessing and analysis using the R software package “Seurat” (v 4.2.0). The conversion of scRNA-seq data into Seurat objects was carried out to ensure the retention of high-quality data. To achieve this, raw matrices were filtered to eliminate cells with transcript counts below 200 and those with mitochondria genes constituting more than 15% of the total genes. Similarly, genes with expression detected in less than three cells were also removed. The “NormalizeData” function from the “Seurat” package was then employed to normalize the data, utilizing the “LogNormalize” method. Principal component analysis (PCA) was used to select the top 20 principal components (PCs) according to the results obtained from the JackStraw and ElbowPlot functions of the Seurat package in R software (v 4.2.1). The “FindNeighbors” and “FindClusters” functions of the Seurat package were used for cell cluster analysis. Subsequently, cells were clustered and visualized using the “RunTSNE” and “RunUMAP” functions. The “FindAllMarkers” function was applied to identify difference-expressed genes between one cluster and all other clusters, employing a |log2 (fold change)| threshold of >1 and an adjusted *p*-value threshold of <0.05. Marker genes for each cluster were found based on these criteria. Finally, cell cluster annotation was accomplished by referencing marker genes from the CellMarker 2.0 databases [22] and genes reported in the relevant literature.

### 2.4. Differentially Expressed Genes (DEGs) Analysis and Signaling Pathways Investigation

Differential gene expression analysis and signaling pathway investigation were conducted to elucidate the molecular mechanisms involved. The “edgeR” package was employed to identify DEGs in both MICA-positive (MICA+) and MICA-negative (MICA−) HCC cells, as well as in macrophages with high MMP9 expression (MMP9+) and low MMP9 expression (MMP9−). The selection criteria for DEGs were set at |log2 fold change (FC)| > 0.5 and a false discovery rate (FDR) < 0.05. Volcano plots depicting the DEGs were generated using the “EnhancedVolcano” R package. Furthermore, gene set enrichment analysis (GSEA) was performed utilizing the 50 hallmark gene sets available in the MSigDB databases. This analysis aimed to confirm the pathways that were either induced or repressed in different experimental groups. The GSEA software (version 4.3.1) was employed to calculate the enrichment score for each gene set, with screening criteria set at |normalized enrichment score (NES)| > 1, nominal *p*-value < 0.05, and FDR q-value < 0.05.

### 2.5. Cell-Cell Communication Analysis and Protein-Protein Interaction (PPI) Network Construction

Cell-cell communication analysis and PPI network construction were conducted to investigate the interactions between various cell types. The “Cellchat” package (v 0.0.2; R package) [23] was utilized to explore these interactions. The CellChat results provided insights into cellular communication patterns through assessing the expression of ligand-receptor pairs. Furthermore, we performed a comprehensive analysis of differential ligand-receptor pairs between different phenotypes of cancer cells and macrophages. To obtain PPI relationships for DEGs within the same signaling pathways, we applied the search tool for the retrieval of interacting genes (STRING, https://stringdb.org/ (accessed on 3 August 2023)) database. Subsequently, the PPI network was visualized using Cytoscape (version 2.1.6).

### 2.6. Single-Cell Data Trajectory Analysis

Single-cell data trajectory analysis was conducted to investigate the differentiation patterns of macrophage subpopulations using the “monocle” R package. Initially, the macrophages were positioned along an inferred trajectory. Subsequently, the reversed graph embedding algorithm of Monocle was employed to shape the trajectory by assigning genes that passed quality control. The data underwent dimensionality reduction and the cells were ordered in pseudotime by Monocle. This analysis facilitated the identification of differentiation trajectories and key genes associated with these trajectories. Macrophages were classified according to CD86 and CD206 expression [24].

### 2.7. Immune Cell Infiltration and Kaplan-Meier (K-M) Curve Evaluation

The levels of immune cell infiltration, specifically 22 types, were analyzed using the CIBERSORT algorithm [25]. The Wilcoxon test was employed and the results were visualized. Survival analysis was conducted using the Kaplan-Meier method, and the statistical significance of differences was determined using the log-rank test. The prognostic predictive accuracy of genes and cells was evaluated by measuring the time-dependent receiver operating characteristic (ROC) curves and calculating the area under the curve (AUC).

### 2.8. Cell Line and Reagents

Liver cancer cell line Huh-7, HepG2 and Hepa1-6, and human monocytic leukemia cell THP-1 (Chinese Academy of Sciences, Shanghai, China) were maintained in DMEM (Gibco, New York, NY, USA) and RPMI-1640 (Gibco, New York, NY, USA), respectively, contained with 100 µg/mL streptomycin, 100 U/mL penicillin (Invitrogen, Waltham, MA, USA) and 10% fetal bovine serum (FBS) (Sigma, Clayton, Australia). The trypsin (Gibco, New York, NY, USA) was used during cell passage. In certain experiments, the cells were treated with phorbol-12-myristate-13-acetate (PMA) (MedChemExpress, Monmouth Junction, NJ, USA, catalog no. HY-18739/CS-6053), human recombinant Pros1 protein (R&D Systems, Minneapolis, MN, USA, catalog no. 9489-PS) or cabozantinib (MedChemExpress, Monmouth Junction, NJ, USA, catalog no. HY-13016).

### 2.9. THP-1 Differentiation, Treatment and Cell Co-Culture

To induce differentiation of THP-1 cells into macrophages, a seeding density of 1 × 10^5^ cells/well was used in 6-well plates, followed by treatment with 300 nM PMA for 24 h. Subsequently, the medium was replaced with fresh RPMI 1640 medium to support macrophage culture. For treatment experiments, macrophages were exposed to 2 μg/mL of human recombinant Pros1 protein for 72 h. Co-culture assays were performed using Transwell chambers (0.4 μm, Corning Inc., Kenneburg, ME, USA), where the bottom chamber contained macrophages and the upper chamber was seeded with Huh-7 cells (1 × 10^5^ cells/well) suspended in DMEM medium. The co-culture system was maintained in a 5% CO_2_, 37 °C incubator for 72 h. Additionally, 5 μM cabozantinib was added to the macrophages stimulated with recombinant human Pros1 protein and the co-cultured macrophages. Following the experimental procedures, total RNA and proteins were extracted from the macrophages for subsequent analyses. We configured the conditioned medium and the normal medium in a 1:1 ratio to meet the necessary nutrients of the cells [26].

### 2.10. Immunofluorescent (IF) Staining and Enzyme-Linked Immunosorbent Assay (ELISA)

Huh-7 cells were seeded in 6-well plates and allowed to adhere for 24 h. Subsequently, the cells were washed three times with phosphate-buffered saline (PBS) and fixed using a 4% paraformaldehyde solution for 15 min. To facilitate permeabilization, the cells were treated with 0.5% Triton x-100 for 20 min at room temperature. Following permeabilization, the cells were incubated overnight at 4 °C with a primary PROS1 antibody (ab280885, abcam, Burlingame, CA, USA). Subsequently, the cells were incubated with Alexa Fluor 594 anti-rabbit antibody (CST, Danfoss, MA, USA) at 37 °C for 1 h. After washing with PBS, the slides were stained with 4′,6-diaminine-2′-phenylindole dihydrochloride (DAPI) and observed using an Olympus Fluview FV1000 III microscope (Olympus, Tokyo, Japan). To determine the levels of soluble MMP9 and MICA in the culture medium, an ELISA kit (mlbio, Shanghai, China) was utilized according to the manufacturer’s specification. The optical density of the samples was measured at 450 nm using a microplate reader.

### 2.11. Immunohistochemistry (IHC) Staining

Each tissue specimen collected in this study underwent formalin fixation and paraffin embedding. Subsequently, the tumor and adjacent liver tissue were co-embedded in the same block. Consecutive sections, measuring 5 microns in thickness, were then cut from each block and mounted onto glass slides. Following dewaxing, rehydration, and antigen retrieval, the anti-AXL (13196-1-AP), MMP9 (10375-2-AP) (Proteintech, Wuhan, China), anti-MICA (BS-0832R, BioSS, Beijing, China), and anti-PROS1 (ab280885, abcam, Burlingame, CA, USA) antibodies were applied to the sections in accordance with standardized institutional protocols. All slides were counterstained with hematoxylin, dehydrated, and sealed using neutral resin.

### 2.12. Lentivirus Infection

A lentiviral vector containing human IRF1 cDNA, MICA cDNA, or an empty vector was generated by Genechem (Shanghai, China). Huh-7 cells were seeded following standardized protocols and infected with lentivirus for 48 h the next day, using the experimental MOI. Subsequently, the culture medium was replaced, and the cells expressing the desired gene were sorted and expanded using flow cytometry. Total RNA and cell lysate protein were separately extracted for subsequent experiments.

### 2.13. Real-Time RT-PCR

Total RNA was extracted using TRIzol reagent and subsequently reverse transcribed into single-stranded cDNA using PrimeScript^TM^ RT Master Mix (Takara Bio, Kyoto, Japan). Real-time quantitative polymerase chain reaction (PCR) was conducted using the SYBR Premix Kit (Takara Bio, Kyoto, Japan). The reaction volume for the all-in-one qPCR Mix was 20 µL, comprising 1 µL of cDNA, 10 µL of 2× all-in-one qPCR Mix, 1 µL of 2 mmol/L reverse primer, 1 µL of 2 mmol/L forward primer, and 6 µL of nuclease-free water. The primer sequences utilized are provided in Appendix A. The relative changes in gene expression were normalized to GAPDH mRNA and determined using the 2^−∆∆Ct^ method.

### 2.14. Western Blot

Cells were collected after intervention, and lysed on ice with high-efficiency RIPA (Solarbio, Beijing, China) for 30 min, then centrifuged at 12,000 R/min for 15 min at 4 °C. The cell debris was removed, the supernatant was collected, and the protein content was determined using the BCA protein assay kit (P0012, Beyotime, Shanghai, China). The protein was separated with SDS-PAGE and transferred onto a 0.45 µm PVDF membrane at low temperatures; the membrane was then blocked with 5% skimmed milk for 1 h. The membrane was incubated with primary antibodies overnight at 4 °C. The primary antibodies applied to western blot analysis contained anti-IRF1 (11335-1-AP), MICA (12619-1-AP), and MMP9 (10375-2-AP) from Proteintech (Wuhan, China), anti-PROS1 (ab280885, abcam, Burlingame, CA, USA), and anti-GAPDH (CST, Danfoss, MA, USA). The membranes were incubated for 1 h with fluorescent secondary antibodies at a 1:20,000 dilution and subsequently an Odyssey imaging system (Odyssey CLX, LI-COR, Lincoln, NC, USA) was used to visualize the result. The acquired bands intensities were quantified using Image J software (version 1.8.0) with the formula: relative abundance (RA) = (target band intensity)/(GAPDH band intensity).

### 2.15. Statistical Analysis

All statistical analyses were conducted using the R software version 4.3.0 (http://www.R-project.org (accessed on 3 July 2023). Unless otherwise noted, *p* < 0.05 was considered as statistical significance.

## 3. Results

### 3.1. The Expression Levels of MMP9 and MICA Were Found to Be Significantly Elevated in Macrophages and Tumor Cells, Respectively, as Observed in the Single-Cell Atlas of Advanced HCC

In order to make the cells finely classified, the clustering analyses were performed using a threshold of 10 resolutions, resulting in the acquisition of 78 distinct clusters. Subsequently, the scRNA-seq data was visualized and the distribution of cell clusters in the multi-dimensional HCC patient tumor and adjacent non-tumor liver was examined using uniform manifold approximation and projection (UMAP) and t-SNE techniques. The identification of cell clusters was accomplished by annotating them based on marker genes sourced from the “Cellmarker” database and relevant references (Figure 1a). Notably, the tumor exhibited a reduction in the number of NK cells, T cells, and B cells, while the proportion of macrophages increased in comparison to the adjacent non-tumor liver tissue (Figure 1b), which was consistent with our previous research [9]. This observation suggested that immunosuppressive cells migrated and accumulated within the tumor microenvironment as the tumor progressed towards a higher malignancy state. Furthermore, our investigation revealed a notable upregulation of several matrix metalloproteinases (MMPs) in tumor-infiltrating macrophages, including MMP9, MMP12, MMP19, ADAM8, ADAM9, and ADAM17. Among these, MMP9 exhibited the most substantial increase in expression levels (Figure 1c,d,f,g). Additionally, we observed a significant upregulation of MICA in tumor cells (Figure 1e,g).

To further elucidate the intricate interplay between MICA+ HCC cells and MMP9+ macrophages, we conducted a meticulous analysis of scRNA-seq data derived from tumor samples. Following data normalization, the top 2000 highly variable genes were selected (Appendix A). Subsequently, we employed the find integration anchors function to integrate two distinct single-cell transcriptome sequencing datasets obtained from tumor and adjacent non-tumor liver samples, based on the highly variable genes. Although a merged dataset was generated through this integration process, it was unable to effectively eliminate the batch effect. Therefore, we opted to utilize the non-merged data, as it minimized the potential errors arising from variations across different experimental batches. Following this, PCA and UMAP dimensionality reduction techniques were applied to the dataset, and the outcomes were visually represented through a heat map, JackstrawPlot, and ElbowPlot (Appendix A).

The tumor cells were stratified into two distinct groups based on their MICA expression levels: the MICA high expression group (comprising MICA+ malignant cells) and the MICA low expression group (comprising MICA− malignant cells). Similarly, the macrophages were categorized into two groups based on their MMP9 expression levels: the MMP9 high expression group (consisting of MMP9+ macrophages) and the MMP9 low expression group (comprising MMP9− macrophages). Identification and classification of cell subpopulations were accomplished via leveraging marker genes obtained from the “Cellmarker” database and relevant references. The marker genes of each cell cluster are provided in Appendix A. In total, we identified and delineated 12 distinct clusters, namely NK cell, CD4 T cell, CD8 T cell, B cell, dendritic cell, MMP9+ macrophage, MMP9− macrophage, MICA+ malignant cell, MICA− malignant cell, endothelial cell, hepatic stellate cell, and epithelial cell (Figure 2a). Some typical marker genes for major clusters are also depicted (Figure 2b,c).

### 3.2. The Activation of the IFN-γ Pathway was Observed in MICA+ Tumor Cells and MMP9+ Macrophages

In order to investigate the differential expression of genes in MICA+ tumors and MMP9+ macrophages, we initially compared MICA+ HCC cells with MICA− HCC cells (Figure 3a), resulting in the identification of 53 downregulated genes and 5391 upregulated genes, as depicted in the volcano plot (Figure 3b). Subsequently, we compared MMP9+ macrophages with MMP9− macrophages (Figure 3c), which revealed 27 downregulated genes and 1362 upregulated genes, as illustrated in the volcano plot (Figure 3d).

To further explore the pathway enrichment associated with the regulation of tumor immunity by MICA and MMP9, we performed GSEA using the all differentially expressed genes (FDR q-value < 0.05) obtained from the previous analyses in MICA+ HCC cells and MMP9+ macrophages. We presented the main enriched pathways separately for MICA+ HCC cells and MMP9+ macrophages (Appendix A). Interestingly, the GSEA enrichment analysis indicated a close association between MICA expression in HCC cells and MMP9 expression in macrophages with the IFN-γ signaling pathway (Figure 3e,f, Appendix A).

### 3.3. The Interaction between MICA+ Tumor Cells and MMP9+ Macrophages was Mediated through the PROS1-AXL and CCL15-CCR1 Pathways

To investigate the interplay between MICA+ tumor cells and MMP9+ macrophages, we employed CellChat, a computational tool, to unravel the intricate cell-cell communication and predict significant biological insights from scRNA-seq data. Given that the tumor microenvironment primarily comprises close interactions between tumor and immune cells, our initial focus was on analyzing the cellular associations within the tumor tissue. By constructing an aggregated cell-cell communication network, we visualized the intensity and frequency of interactions across all cell types. Notably, the network analysis revealed that MICA+ HCC cells exhibited the highest degree of interactions with other cell types (Figure 4a,b).

Next, we employed CellChat to conduct pathway-based analyses of ligand-receptor (L-R) interactions between MICA+/MICA− tumor cells and MMP9+/MMP9− macrophages. Notably, we observed the presence of L-R interactions involving PROS1-AXL and CCL15-CCR1 between MICA+ tumor cells and MMP9+ macrophages (Figure 4c). To further validate these findings, a violin plot of the aforementioned genes in the CellChat analyses confirmed that PROS1 and CCL15 were exclusively expressed in MICA+ tumor cells, while AXL and CCR1 were exclusively expressed in MMP9+ macrophages (Figure 4d). Moreover, we utilized STRING and GSEA analyses to construct protein-protein interaction (PPI) networks, which revealed a close association between PROS1, AXL, CCL15, CCR1, MICA, MMP9, and the IFN-γ response pathway (Figure 4e).

### 3.4. The Predominant Expression of MMP9 Was Observed in M2-Like Macrophages

We subsequently elucidated the phenotype and functional characteristics of macrophages expressing MMP9. To achieve this, we employed the “Monocle” approach to identify the dynamic polarization of macrophages within the TME. Our analysis categorized macrophages into five distinct groups based on their expression of MMP9, CD86, and CD206: MMP9−CD86−CD206−, MMP9−CD86+CD206−, MMP9−CD86+CD206+, MMP9+CD86+CD206−, and MMP9+CD86+CD206+. Notably, all cells were projected onto a single root and two branches, revealing an intriguing pattern. Specifically, MMP9+ macrophages were predominantly located at the terminal end of the differentiation pathway, while MMP9− macrophages were predominantly found at the initiation stage (Figure 5a).

Furthermore, pseudotime analysis demonstrated that the tree structure originated from M0-like cells characterized by CD68+CD86-CD206−, followed by M1-like cells marked by CD68+CD86+CD206−, and M2-like cells marked by CD68+CD86−CD206+ (Figure 5b). Importantly, MMP9 exhibited significant expression in M2-like macrophages. Additionally, our investigation revealed a positive correlation between AXL and CCR1 with M2-type macrophages in the TCGA dataset (Figure 5c). Collectively, our findings indicate that MMP9+ macrophages possess a phenotype and function consistent with M2-like macrophages.

### 3.5. IRF1 Induced the UpRegulation of PROS1 in HCC Cells and Also Enhanced the Expression of PROS1 Specifically in MICA+ HCC Cells

Given the pronounced expression of PROS1 and AXL compared to CCL15 and CCR1 in HCC (Figure 4d), we proceeded to focus on investigating the PROS1-AXL pathway for experimental validation. Initially, we assessed the mRNA and protein expression levels of PROS1 and AXL across various human tumor types using the Human Protein Atlas (HPA) database. Notably, IHC staining revealed medium/high PROS1 and AXL expression in liver cancers (Appendix A). Furthermore, we observed high PROS1 mRNA expression in Huh7 cells rather than immune cells, while high AXL mRNA expression was detected in some immune cells rather than Huh7 cells (Appendix A). Additionally, analysis of TCGA database confirmed the upregulation of MICA and MMP9 expression in HCC (Appendix A).

In line with the aforementioned findings, our IHC staining results provided further validation by demonstrating elevated protein expression levels of PROS1, AXL, MICA, and MMP9 in HCC tissues compared to normal liver tissues (Figure 6a). Moreover, scRNA-seq analyses revealed a significant increase in PROS1 expression in HCC tumor cells exhibiting positive expression of IRF1 or MICA (Figure 6b,c). This observation suggested a positive correlation between IRF1 and MICA expressions and PROS1 in HCC cells.

Building upon our previous findings that demonstrated the role of DNA damage in inducing MICA expression via IRF1 at the transcriptional level in HCC cells [9], we proceeded to investigate whether IRF1 acts as an upstream regulator of PROS1. To this end, we employed lentiviral IRF1 cDNA transduction to upregulate IRF1 expression in Huh-7 cells. As anticipated, the mRNA and protein levels of PROS1 exhibited a significant increase in IRF1+ Huh-7 cells (Figure 6d,e). Additionally, lentiviral transduction-mediated overexpression of MICA in Huh-7 cells resulted in a notable elevation in both PROS1 mRNA and protein levels (Figure 6f,g). Consistently, immunofluorescence staining revealed an upregulation of PROS1 protein expression in Huh-7 cells transfected with IRF1 and MICA overexpression (Figure 6h,i).

### 3.6. Stimulation of MICA+ HCC Cells Resulted in the Secretion of MMP9 through the PROS1-AXL Axis, Thereby Facilitating the Proteolytic Shedding of MICA into Soluble MICA (sMICA)

Since the CellChat analyses indicated the interaction between MICA+ HCC cells and MMP9+ macrophages through the PROS1-AXL axis (Figure 4c), we proceeded to validate the existence of this axis. Subsequently, we observed a significant increase in both MMP9 mRNA and protein expression levels in macrophages upon stimulation with recombinant PROS1 protein, particularly after 72 h (Figure 7a,b). Additionally, we noted a significant increase in AXL mRNA expression and a significant increase in protein levels in macrophages induced via recombinant PROS1 protein (Figure 7a,b). These findings suggested that the upregulation of MMP9 in TAMs occurs via a PROS1-AXL-dependent pathway. Furthermore, we observed a noticeable upregulation of CD163 and CD206, marker associated with M2-type macrophages (Figure 7a), which aligns with our previous analyses of single-cell and TCGA data, indicating that MMP9+ macrophages may exhibit a propensity towards an M2-like phenotype (Figure 5c).

Furthermore, an ELISA assay was conducted to determine the MMP9 levels in the culture medium of macrophages induced via the recombinant PROS1 protein, revealing significantly higher levels compared to the negative control group (*p* = 0.009) (Figure 7c). The culture medium from the two groups was extracted and designated as the MMP9+ group and MMP9− group. Subsequently, MICA+ Huh7 cells were cultured using a mixed medium consisting of MMP9+ or MMP9− group conditioned medium and normal medium in a 1:1 ratio. Interestingly, the soluble MICA level in the medium of MICA+ Huh7 cells cultured with conditioned medium from the MMP9+ group was found to be significantly higher than that of the MMP9- group (*p* = 0.026) (Figure 7d).

Given the activation of the IFN-γ pathway in MICA+ tumor cells (Figure 3e,f) and the role of IRF1 as a key transcription factor in this pathway, which is also an upstream regulator of PROS1 (Figure 6b,d,e,h), along with the observed increase in PROS1 expression coinciding with MICA expression in HCC cells (Figure 6c,f,g,i), our study aimed to investigate whether IRF1+ and MICA+ HCC cells induce MMP9 expression in macrophages through a cellular co-culture model. To achieve this, we co-cultured IRF1+ and MICA+ Huh7 cells with macrophages and observed an upregulation of MMP9 mRNA and protein in the macrophages (Figure 7e,f). To further validate the involvement of the PROS1-AXL axis in the upregulation of macrophage MMP9 via MICA+ tumor cells, we introduced the AXL inhibitor cabozantinib to macrophages stimulated via recombinant human PROS1 protein and co-cultured with IRF1+ Huh7 cells. Remarkably, the inhibition of macrophage AXL resulted in the reversal of the PROS1-induced upregulation of macrophage MMP9 (Figure 7g,h).

## 4. Discussion

In this study, we conducted a comprehensive analysis at the single-cell level to characterize the cellular composition of both cancerous and non-cancerous liver tissues in patients with advanced hepatitis B-associated HCC. Specifically, we explored the potential interaction between MICA+ tumor cells and MMP9+ macrophages. Subsequently, experimental validation confirmed that MICA+ tumor cells upregulated MMP9 expression in macrophages through the PROS1-AXL axis. Additionally, we confirmed that stimulation of PROS1 results in the transformation of macrophages into the M2-like phenotype. Finally, we observed that MMP9 secreted by macrophages hydrolyzed MICA, leading to the release of sMICA and promoted immune evasion by the tumor (Figure 8). These findings certified that AXL in macrophages may serve as a potential therapeutic target for HCC.

Currently, immune checkpoint blockade (ICB) therapy is widely acknowledged as a significant advancement in the field of systemic therapy for HCC. Nevertheless, the TME plays a crucial role in facilitating tumor immune evasion, as well as the development of primary and secondary resistance, post-therapeutic tumor hyper-progression, and associated adverse effects, thereby presenting formidable challenges to the efficacy of ICB treatment [27,28]. To address these challenges, the combination of ICB with molecular therapy, chemotherapy, or radiotherapy has emerged as a promising approach. Notably, the induction of DNA damage response (DDR) in HCC tumors via these combined therapies has been shown to activate anti-tumor immunity and remodel the TME, thereby enhancing the effectiveness of ICB therapy. However, both cancerous and non-cancerous cells within the TME have demonstrated the ability to adapt to the microenvironmental changes induced by DNA damage, thereby developing resistance to the therapy [29].

In our previous investigation, we observed that DNA damage-induced upregulation of MICA expression resulted in the activation of NKG2D-mediated anti-tumor immune responses by NK cells and CD8+ T cells [9]. Conversely, several studies have reported that elevated MICA expression in HCC and small cell lung cancer (SCLC) is associated with an unfavorable prognosis [30,31]. This association may be attributed to the hydrolytic shedding of MICA mediated by the MMPs. Notably, the shedding of NKG2D ligands has been validated in some cancers [31,32]. Furthermore, we observed a negative correlation between the level of soluble NKG2D ligands and the infiltration of memory T cells, while a positive correlation was found with the infiltration of CD163+CD206+ macrophages, which are known to promote tumor growth [33]. Additionally, our study also confirmed the prominent MMP9 upregulation in these M2-like macrophages (Figure 7a).

To elucidate the underlying mechanism of MICA+ tumor cells inducing MMP9 upregulation in macrophages and subsequent MICA shedding, further investigation is warranted. We employed scRNA-seq analyses to investigate the communication between MICA+ tumor cells and MMP9+ macrophages via the PROS1-AXL axis and revealed an upregulation of the IFN-γ signaling pathway in MICA+ tumor cells. Our previous studies have demonstrated that IFN-γ can upregulate the expression of IRF-1 in HCC cells [6], while DNA damage can induce MICA expression in HCC cells through IRF1 [9]. Consistent with our previous research, we observed a significant upregulation of PROS1 in MICA+ and IRF1+ tumor cells (Figure 6d–i). Based on these observations, we hypothesized that IRF1 may simultaneously upregulate the expression of both MICA and PROS1. To support this hypothesis, we utilized data from the ENCODE project (https://www.encodeproject.org) and identified IRF1 as a transcription factor for PROS1.

PROS1 and Gas6 have been identified as ligands for the Tyro3, AXL, and Mer (TAM) families of tyrosine kinase receptors [16]. Previous studies have demonstrated that Gas6 binds to all three TAM receptors, while PROS1 activates Tyro3 (with high affinity) and Mer, but not AXL [34]. However, recent studies have provided evidence that PROS1 can also activate AXL in tumor cells, thereby promoting tumor cell proliferation and invasiveness [21,35,36]. PROS1 and TAM receptors signaling pathways have been recognized as a crucial negative regulator of the immune system, playing a role in limiting the intensity and duration of immune responses [17,37]. Furthermore, AXL has been found to promote the transition of macrophages from an anti-tumorigenic M1-like phenotype to a tumorigenic M2-like phenotype [38], and it has been implicated in inducing sorafenib resistance in HCC by increasing the population of M2-like macrophages [39]. Moreover, there is evidence suggesting that activation of the AXL signaling pathway can upregulate the expression of MMP9 [40,41]. Consistent with the research, HCC cells were found to secrete PROS1, which subsequently upregulated the expression of MMP9 in macrophages through the activation of AXL receptors (Figure 7a,b,e,f). Finally, we observed that MMP9 secreted by macrophages hydrolyzed MICA, leading to the release of sMICA and promoting immune evasion by the tumor (Figure 7c,d).

Cabozantinib, a recently approved second-line treatment for malignancies, has demonstrated inhibitory effects on various receptor tyrosine kinases, including VEGFR, MET, and AXL [42]. Notably, approximately 40% of HCC patients exhibit elevated AXL expression, which has been associated with vascular invasion and poor survival outcomes [40]. The therapeutic application of cabozantinib in cancer involves the inhibition of AXL tyrosine kinase activity and the subsequent blockade of downstream pathway activation [43]. In a study of 707 patients with advanced HCC who had previously been treated with sorafenib, cabozantinib significantly extended overall survival (10.2 vs 8.0 months) and progression-free survival (5.2 vs. 1.9 months) compared to placebo [42], which similar results were found in another Phase 3 CELESTIAL trial [44]. Furthermore, in a study of cabozantinib combined with nivolumab as a neoadjuvant therapy for unresectable HCC, 12 patients demonstrated a major/complete pathologic response rate of 42% (5/12) postoperatively, and the study has also shown that cabozantinib and nivolumab produced a therapeutic response by modulating tumor-associated macrophages [45]. Interestingly, patients who developed bone metastases after hepatectomy also benefited from cabozantinib combined with nivolumab, with progression-free survival of more than 25 months [46]. Hence, our findings elucidated a novel mechanism whereby the PROS1-AXL axis induced the secretion of MMP9 from macrophages, thereby providing valuable insights and supporting the therapeutic application of AXL inhibition in HCC.

## 5. Conclusions

MICA+ HCC cells were found to secrete PROS1, which subsequently upregulated the expression of MMP9 in macrophages through the activation of AXL receptors. This heightened MMP9 activity led to the proteolytic shedding of MICA molecules. Consequently, the released soluble MICA (sMICA) played a pivotal role in promoting tumor immune evasion.

## Figures and Tables

**Figure 1 cancers-16-00269-f001:**
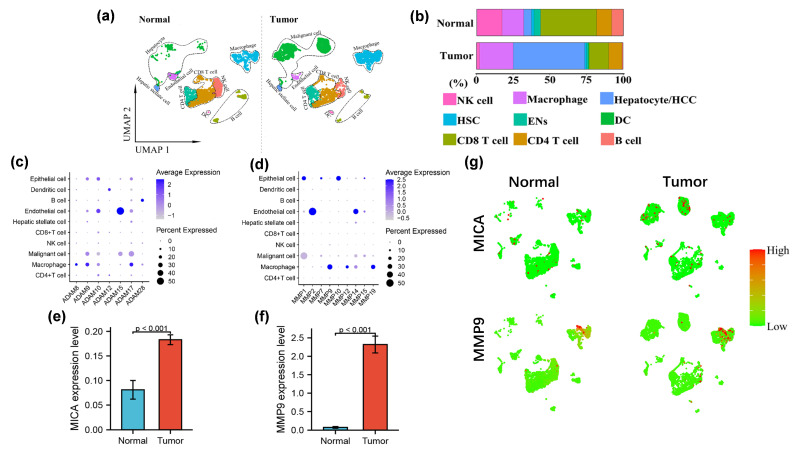
MMP9 and MICA expression on single cell atlas of advanced HCC. (**a**) UMAP plots of 4979 cells from no-tumor and 5359 cells from tumor tissue are shown for 9 clusters in each plot. Each cluster is illustrated by a different color. (**b**) Proportions of the various types of cells in the tumor and non-tumor samples. (**c**,**d**) Dot plots of MMPs and ADAMs expression are shown in each cell cluster. The size of the dot indicates the percentage of the gene expression in each cluster. The depth of color is related to the intensity of gene expression. (**e**,**f**) The statistical analysis of MICA and MMP9 expression level in HCC tumor and non-tumor tissue. (**g**)The expression of MICA and MMP9 in tumor and non-tumor profiled on the UMAP plots.

**Figure 2 cancers-16-00269-f002:**
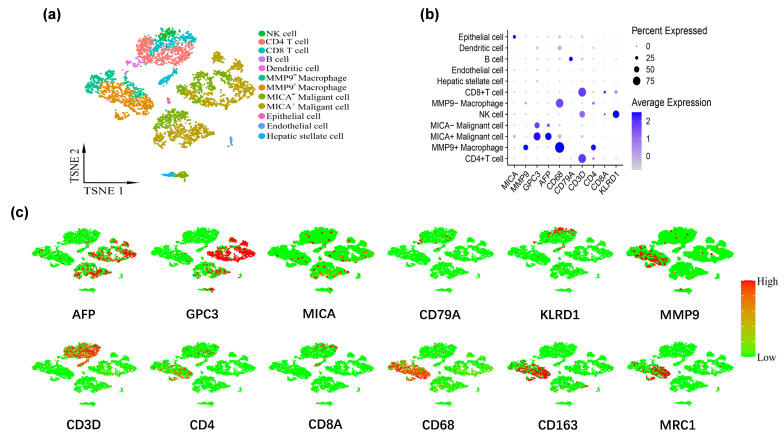
MMP9 and MICA expression on single cell atlas of advanced HCC respectively significantly increased in macrophages and tumor cells. (**a**) UMAP plots of 5359 cells from tumor tissue. The tumor cells and macrophages were further grouped according to the expression levels of MICA and MMP9, respectively. (**b**) Dot plot of marker genes expression in each cluster. The size of the dot indicates the percentage of the gene expression in each cluster. The depth of color is related to the intensity of gene expression. (**c**) The feature map depicts single-cell gene expression of individual marker genes.

**Figure 3 cancers-16-00269-f003:**
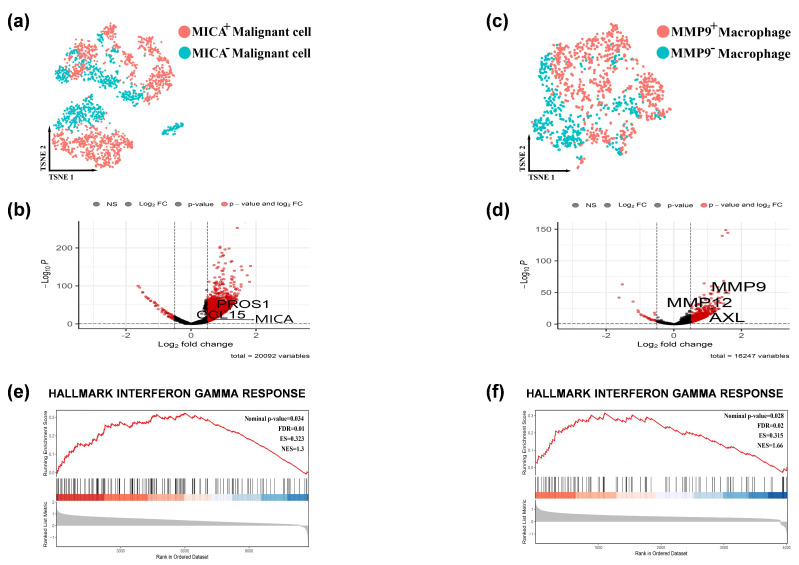
The IFN-γ pathway was activated in the MICA+ tumor cells and MMP9+ macrophages. (**a**) Feature plots depicting single-cell tumor cell expression of MICA. (**b**) Volcano plots showing the single-cell transcriptional profile difference between MICA+ and MICA− tumor cells. (**c**) Feature plots depicting single-cell macrophage expression of MMP9. (**d**) Volcano plots showing the single-cell transcriptional profile difference between MMP9+ and MMP9- macrophage. (**e**,**f**) GSEA analysis revealed single-cell MICA and MMP9 expression involved in the IFN-γ signaling pathway. Red indicated that gene was highly expressed in the MICA+ tumor cells/ MMP9+ macrophage, and blue indicated that gene was highly expressed in the MICA− tumor cells/ MMP9− macrophage.

**Figure 4 cancers-16-00269-f004:**
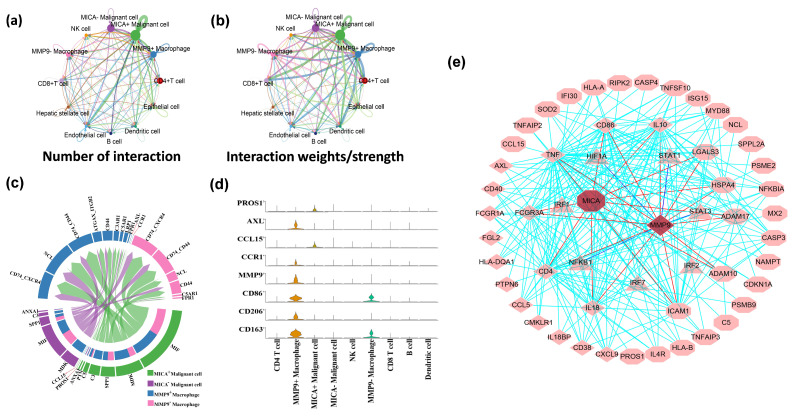
MICA+ tumor cells communicated with MMP9+ macrophages via the PROS1-AXL and CCL15-CCR1 pathways. (**a**,**b**) The interaction net count and interaction weight plot of HCC cells. The thicker line indicates the higher number of interactions and the stronger weights/strength of interactions between the two cell types. (**c**) Chord diagram showing all the significant interactions (ligand-receptor pairs) between MICA+/MICA− tumor cells and MMP9+/MMP9− macrophage. (**d**) The violin plot showing PROS1-AXL/CCL15-CCR1 (ligand-receptor pairs) and some macrophage marker genes expression in different cell clusters. (**e**) PROS1-AXL/CCL15-CCR1 ligand-receptor pairs and some major IFN-γ signaling pathway genes inside tumor cells and macrophage. Diamonds represent genes that are upregulated in macrophages. Octagons represent genes that are upregulated in tumor cells. The triangles represent transcription factors.

**Figure 5 cancers-16-00269-f005:**
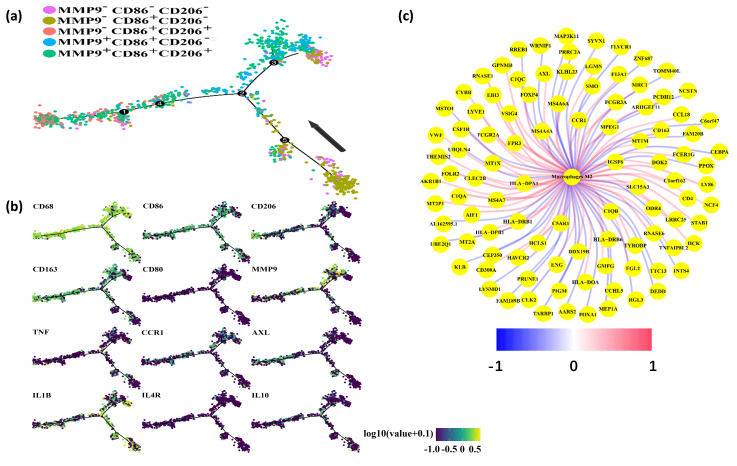
MMP9 mainly expressed in M2-like macrophages. (**a**) Pseudotime analysis explored the development trajectories of macrophage. Each dot corresponds to a single cell, the macrophage subtype is depicted according to specific markers expression level, and arrows indicate the initial position and direction of differentiation. (**b**) Representative gene expression plotted as a function of pseudotime. (**c**) Genes associated with M2-like macrophages in the TCGA dataset. Red represents positive correlation, and blue represents negative correlation.

**Figure 6 cancers-16-00269-f006:**
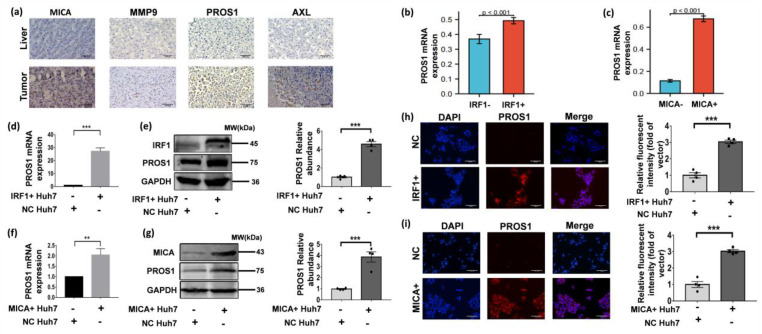
IRF1 upregulated PROS1 in HCC cells and also enhanced PROS1 expression in MICA+ HCC cells. (**a**) The typical IHC images of MICA, MMP9, PROS1, and AXL in non-tumor and tumor live (×400 magnification). Scale bars are 200 µm. (**b**,**c**) PROS1 expression levels of IRF1 and MICA positive tumor cells in scRNA-seq data. (**d**) PROS1 mRNA expression levels of Huh-7 cells infected by lentivirus IRF1 cDNA or empty vector (NC) (50 MOI) for 48 h. (**e**) PROS1 protein level of Huh-7 cells infected by lentivirus IRF1 cDNA or empty vector (NC) (50 MOI) for 48 h. The quantitative analyses are shown on the right. (**f**) PROS1 mRNA expression level of Huh-7 cells infected by lentivirus MICA cDNA or empty vector (NC) (50 MOI) for 48 h. (**g**) PROS1 protein level of Huh-7 cells infected by lentivirus MICA cDNA or empty vector (NC) (50 MOI) for 48 h. The quantitative analyses are shown on the right. (**h**,**i**) The typical IF images show PROS1 protein level (red staining) in Huh-7 cells which were respectively transfected with lentiviral IRF1 and MICA (×200 magnification). Scale bars are 150 µm. Quantification of PROS1 via relative fluorescent intensity is shown. ** *p* < 0.01, *** *p* < 0.001, via two-tailed unpaired *t*-test. The original uncropped western blot images are shown in Appendix A.

**Figure 7 cancers-16-00269-f007:**
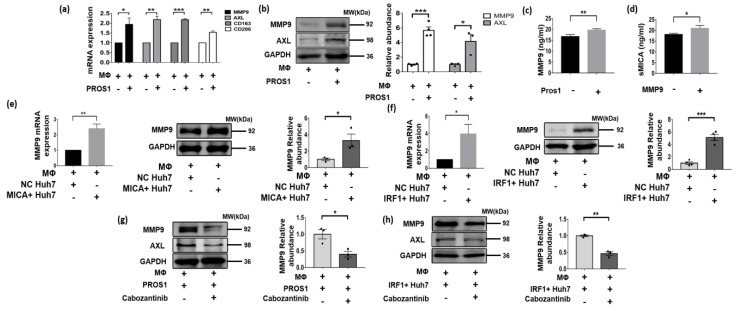
MICA+ HCC cell stimulated MMP9 secretion via PROS1-AXL axis and promoted proteolytic shedding of MICA into soluble MICA (sMICA). (**a**) MMP9, AXL, CD163 and CD206 mRNA expression was detected via qPCR in Huh-7 cells stimulated by recombinant human PROS1 protein 2 μg/mL for 72 h. (**b**) The whole lysate for PROS1 and AXL protein determined via western blot is shown in macrophage stimulated by recombinant human PROS1 protein for 72 h. The quantitative analyses are shown on the right. (**c**,**d**) The level of MMP9 and soluble MICA in the conditioned medium was determined via an ELISA procedure. (**e**) MMP9 mRNA expression was detected via qPCR in macrophage co-cultured with MICA+ and NC Huh7 cells. The whole lysate for MMP9 protein determined via western blot is shown in macrophage cultured with MICA+ and NC Huh7 cells. The quantitative analyses are shown on the right. (**f**) MMP9 mRNA expression was detected via qPCR in macrophage co-cultured with IRF1+ and NC Huh7 cells. The whole lysate for MMP9 protein determined via western blot is shown in macrophage cultured by IRF1+ and NC Huh7 cells. The quantitative analyses are shown on the right. (**g**) MMP9 protein expression was measured via western blot in macrophage stimulated by preconditioned cabozantinib (5 μM) and recombinant human PROS1 protein for 72 h. The quantitative analyses are shown on the right. (**h**) MMP9 protein expression was measured via western blot in macrophage co-cultured with IRF1+ and NC Huh7 cells treated by cabozantinib (5 μM). The quantitative analyses are shown on the right. * *p* < 0.05, ** *p* < 0.01, *** *p* < 0.001, via two-tailed unpaired *t*-test. The original western blots are shown in Appendix A.

**Figure 8 cancers-16-00269-f008:**
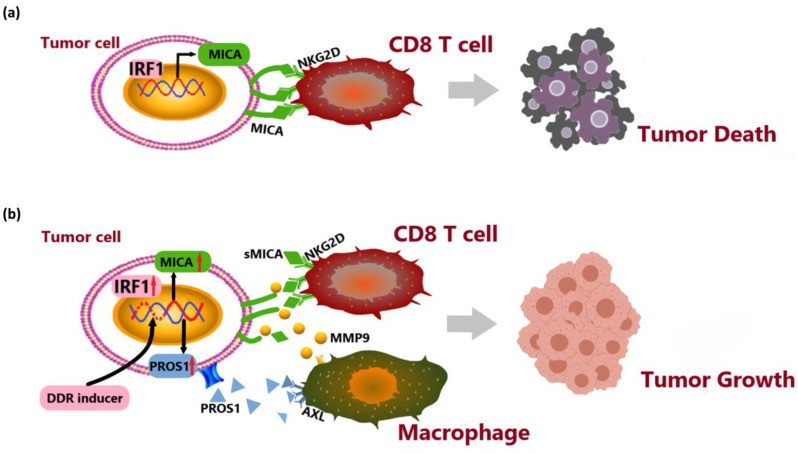
System illustration of MICA+ tumor cell upregulated macrophage-secreted MMP9 via PROS1-AXL axis to induce tumor immune escape in advanced HCC. (**a**) IRF1 transcriptionally upregulated MICA expression to activate NKG2D-mediated anti-tumor immune responses by CD8+ T cells. (**b**) DNA damage induced MICA and PROS1 expression via IRF1. PROS1 activated MMP9 secreted from macrophages via AXL receptor to hydrolyze MICA. The release of soluble MICA (sMICA) promoted immune evasion by the tumor.

## Data Availability

The raw data supporting the conclusions of this article will be made available by the authors without undue reservation.

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
