# Peer review of "MICA+ Tumor Cell Upregulated Macrophage-Secreted MMP9 via PROS1-AXL Axis to Induce Tumor Immune Escape in Advanced Hepatocellular Carcinoma (HCC)"

_cancers, 2024, doi:10.3390/cancers16020269_

Round 1

Reviewer 1 Report

Comments and Suggestions for Authors

The computational part, which is substantial in the manuscript by Wu et al., cannot be judged by this reviewer, nor if the related described methods are enough to reproduce the study.

The study does not address why the accumulation of TAMs associates with poor prognosis in HCC, as the introduction seems to allude while reading.

Please quantify the immune cell population clusters to have a hierarchy of representativeness in the tumor tissues compared to normal ones.

Immunoblot analyses were not performed in FACS-sorted cells, which would have possibly resulted in more informative correlations.

Is there any proof that MMP9 causes the promotion of immune evasion in other cancers?

raw 66, natural killer

Comments on the Quality of English Language

Good.

some typos are present

Reviewer 2 Report

Comments and Suggestions for Authors

Dear authors,

I have carefully reviewed your manuscript entitled “MICA+ Tumor Cell Upregulated Macrophage-Secreted MMP9 2 Via PROS1-AXL Axis To Induce Tumor Immune Escape in 3 Advanced Hepatocellular Carcinoma (HCC)”.

While the study is conceptually intriguing and the conclusions are supported by the data, the manuscript is difficult to read due to its excessive length and frequent shifts between topics. This makes an otherwise interesting study challenging to read and to comprehend. Therefore, I believe that significant revisions on the both manuscript structure and content are necessary before the manuscript can be considered for publication.

I have provided detailed point-by-point feedback below for your consideration and response.

Point-by-point feedback:

The abstract should be significantly simplified and consolidated. You have included too many abbreviations, which cannot be introduced within the abstract. Please rewrite the abstract to convey the main message in a way that can be understood by someone who is not a specialist in this research field. Ideally, the abstract should not exceed 270 words in length.

Similarly, the introduction included in your study is too long. As I recommended for the abstract, I advise you to shorten and focus it. As it stands, it is too dispersed and lacks a clear structure. I had to read it more than once to understand where the challenges lie and which question you were asking. Additionally, I suggest keeping the introduction below 380 words and focusing on information that provides readers with the background necessary to understand the problem your work aims to solve.

Line 71 and 74, essentially repeat the same concept, it would be advisable to consolidate in one simple paragraph.

Line 80, What is the TIMER database? Reference?

Line 99, the author states, 'In this study, we present a comprehensive single-cell atlas of advanced HCC.' In my understanding, 'a comprehensive single-cell atlas' implies coverage of all the different types of cells in advanced HCC. However, this study clearly does not achieve that. Please edit accordingly.

Lines 99 to 108: The paragraphs between these lines essentially summarize your main findings. However, they are poorly written, making it difficult to understand the intended message until the manuscript is read. Therefore, I recommend simplifying the message, focusing on conveying the meaning of your findings without delving into a detailed mechanistic description

Line 133, the reference [15] should be placed next to the GSE149614.

Line 114, the authors stated, 'A total of 10,649 cells and 25,712 genes were obtained from two patients.' The author should modify/simply the text to clarify that this is from an already published study and simply include the reference to the original work.

Line 275 to line 285, this portion of 'Section 3.1' could and should be removed. The authors have downloaded and reanalyzed the raw data, not generated it. If the authors believe that this information must be included in their work, then please relocate it to the supplementary materials and methods section.

Line 286, the authors stated, “Based on the references and debugging effects”. The meaning of this statement is unclear. Please simplify the message, replacing complex descriptions with a brief, straightforward (and referenced) explanation of the type of analysis conducted and its outcome.

Line 339 to line 345, this paragraph is hard to read. The authors should consider replacing it with a “Cellmarker” table.

Line 367, the authors performed GSEA by using the differentially regulated genes. Did the authors used all the differentially regulated genes or the “top 2,000 highly variable genes” as indicated in line 313? Please specify.

Line 425 to line 427, please include appropriate reference(s) supporting the combination of cluster of differentiation markers that represent each type of immune cell.

Line 432 to line 453, this complete paragraph is really difficult to read and understand, it should be much shorter and convey a precise message.

Line 483, the authors wrote “we proceeded to investigate the existence of this axis” in my understanding the authors did not “investigate” but rather “validated” its presence, please revise.

Line 487, the authors wrote “Additionally, we noted a remarkable increase in AXL mRNA”, is this “remarkable” or “significant”?

Line 517, It is not clear to me why the authors diluted conditioned medium and normal medium in a 1:1 ratio to perform this experiment. Please include the description of “why” and move to M&M section.

The discussion is too long and dispersive. The authors should emphasize their key findings and discuss them in this section. They should carefully revise the verbal forms used in this part. I would recommend moving the model from line 546 to the end of the discussion.

Line 602, please include the figure to which this text is referring.

Line 609 to line 611, this can be fully removed as does not provide any additional information

Comments on the Quality of English Language

The manuscript poses challenges in readability. Numerous words, some of which I have highlighted in my point-by-point comments, seem out of context. There is a need for the harmonization of verbal forms throughout the manuscript. Additionally, the overall length of your manuscript appears excessive, exceeding the ideal length for your story by 200 to 300 words. This extended length not only makes the manuscript cumbersome but also hinders the readability of what could otherwise be a conceptually intriguing story. I recommend a meticulous review and revision of the language, ensuring coherence and conciseness to enhance the overall clarity and impact of your work. To improve the readability of your work, consider streamlining the text by eliminating unnecessary words or phrases. I also suggest relocating the section describing methodologies to the material and method section. This adjustment should help bring the overall length closer to the desired target.

Reviewer 3 Report

Comments and Suggestions for Authors

The manuscript represents an interesting analysis of the role of cell activation MICA expression and secretion, and the role of IRF-1 and MMP-9in tumour progression and escape. 

The aim and the experiments were designed accordingly. There are, however, several issues that should be analyzed carefully. The macrophages that secrete MMP9 which type of macrophages are they? Is the secretion of MMP-9 dependent on cell activation? and if so, by whom? The MICA-negative cells are cold tumours? Do they associate with TAM in the same way as the MICA-positive cells?  Does the inhibition of NKfB play a role in the expression? Do the antiinflammatory molecules prevent the expression of MICA or enhance the release? Why use the THP-1 cell line and how do the authors certify they were differentiated properly?

The discussion needs to be rewritten since the possible response to the therapy is crucial in the process. 

Comments on the Quality of English Language

Minor grammatical mistakes were encountered
